# Expanding the Scope of Adenoviral Vectors by Utilizing Novel Tools for Recombination and Vector Rescue

**DOI:** 10.3390/v16050658

**Published:** 2024-04-23

**Authors:** Julian Fischer, Ariana Fedotova, Clara Bühler, Laura Darriba, Sabrina Schreiner, Zsolt Ruzsics

**Affiliations:** Institute of Virology, University Medical Center Freiburg, Medical Faculty, University of Freiburg, 79104 Freiburg, Germany; julian.fischer@uniklinik-freiburg.de (J.F.); ariana.fedotova@uniklinik-freiburg.de (A.F.); sabrina.schreiner-gruber@uniklinik-freiburg.de (S.S.)

**Keywords:** human adenoviruses, viral bacmids, replication-competent vectors, transgene insertion sites, mutagenesis of viral genomes

## Abstract

Recombinant adenoviruses are widely used in clinical and laboratory applications. Despite the wide variety of available sero- and genotypes, only a fraction is utilized in vivo. As adenoviruses are a large group of viruses, displaying many different tropisms, immune epitopes, and replication characteristics, the merits of translating these natural benefits into vector applications are apparent. This translation, however, proves difficult, since while research has investigated the application of these viruses, there are no universally applicable rules in vector design for non-classical adenovirus types. In this paper, we describe a generalized workflow that allows vectorization, rescue, and cloning of all adenoviral species to enable the rapid development of new vector variants. We show this using human and simian adenoviruses, further modifying a selection of them to investigate their gene transfer potential and build potential vector candidates for future applications.

## 1. Introduction

Human adenoviruses (HAdVs) are non-enveloped double-stranded DNA viruses with a genome size of about 36 kilobase pairs (kb). They induce mainly asymptomatic or mild infections. Only a few exceptional types of HAdVs can be associated with severe pathogenesis in immune-competent individuals. Still, adenoviruses are the second most common causative agent inducing infection-related morbidity in immunocompromised patients [1]. Their pathogenic features are meant to be associated with their genetic variability among their structural genes determining tissue tropisms [2] and the immune modulatory E3 region, which shows species-specific organization [3]. In recent decades, vectors developed based on adenoviruses (AdVs) have successfully been approved as platforms for recombinant vaccines and gene therapy [4]. Though more than 100 human adenovirus (HAdV) types have been identified to date, which can be sorted into seven species, only a few types have been studied in molecular detail and, therefore, could be developed into useful vector platforms. To date, more and more data suggest that vector features of specific AdVs can considerably vary and be influenced by type-specific factors [5]. Therefore, finding the optimal solution for the emerging application needs requires expanding the detailed molecular biological studies for as many AdV types as possible. An operational reverse genetic workflow based on infectious genomic clones has been available for HAdVs of species C since the 1990s [6]. It was further developed and applied to model types representing most species of HAdV, allowing the generation of virus mutants and developing vectors by rational design based on specific HAdVs [7,8,9,10]. However, to date, these technologies have not reached high efficiencies, which are available for HAdV-C5-based platforms, and still maintain a lag compared to the technologies available for concurrent vector platforms, especially regarding the library-based applications established for recombinant AAVs [11]. We recently developed a reverse genetic workflow for HAdV-C5-based recombinant AdV vectors, which allowed a scalable library efficiency for constructing modified vector particles [12].

In this study, we tested the applicability of the key technologies of our new workflow for the applicability on various model types from six HAdV species. First, we recombined viral genomic DNA from cultured AdVs with bacmid fragments to generate infectious genomic clones in *E. coli* [13]. Then, we tested the ability to create recombinant AdV particles using our novel rescue methodology, named CRISPR/Cas9 terminal resolution (CTR) [14]. We then tested the applicability of the half-site-dependent fragment recombination (HFR) [12], which allowed the seamless manipulation of rAdV bacmids at library efficiency levels for a selected set of target genomes.

CTR utilizes CRISPR/Cas9-induced double-strand breaks targeted within the bacmid to regions adjacent to viral ITRs, releasing the end of the viral genome from the circular bacmids and allowing viral replication. We proved that it can be applied to rescue HAdV-C5-based first-generation vectors and confirmed that it can be used to rescue HAdV-E4-based rAds [14]. CTR requires the (re)construction of a new bacmid, generating a common targeting site, which theoretically allows the rescue of all rAdVs from their circular bacmids with reasonable efficiency. In addition, a more efficient but type-specific guide RNA could be designed to rescue both HAdV-C5- and HAdV-E4-based bacmid constructs. This principle was also tested here for species A, B, and F.

HFR combines the precision of lambda-red-recombination [15] with the high efficiency of Gibson assembly [16] to permit seamless mutagenesis or insertion at virtually any region within the genome. Once modification targets are chosen, the half-sites of an endonuclease not cutting within the genomic bacmid are chosen. Intermittent regions are then replaced with antibiotic resistance via lambda-red recombination. Alongside antibiotic resistance, half sites are also completed, allowing the enzymatic release of the cassette mentioned above. Linear DNA fragments (such as PCR products, enzymatically released fragments, or synthetic DNA) are then used to restore original sequences and target modification via Gibson assembly. In this paper, we tested genetically modified species B and E-derived replication-competent vectors for several insertion sites for transgene expression as practical examples of the applicability of HFR. In addition, we also characterized the fitness and gene expression profiles of the viable insertion sites in the context of both B- and E-derived vectors.

## 2. Materials and Methods

### 2.1. Cells, Viruses, and Bacterial Strains

The A549 (ATCC, Manassas, VA, USA, #CCL-185), 293A (Thermo Fisher Scientific, Waltham, MA, USA, #R70507), and HeLa (ATCC, Manassas, VA, USA, #CCL-2) cells used in this study were cultured using Dulbecco’s modified Eagle’s medium (DMEM) supplemented with 10% fetal calf serum (FCS) and 100 U/mL penicillin–streptomycin (P/S). The original viruses used in this study are listed in Table 1. Bacmids containing recombinant viral genomes were maintained in *E. coli* NEB10beta. Transfer plasmids were maintained in *E. coli* Pir1.

### 2.2. Construction of Viral Bacmids

Viral bacmids were generated as described previously [14] by combining viral DNA from cultured virus with PCR-derived fragments of pKSB2 vector backbone [17]. We used HFR, as described previously, to generate recombinants expressing reporter transgenes [12]. Briefly, after identifying target insertion sites, a kanamycin resistance cassette was used for selection and inserted between the half sites of a natively non-cutting endonuclease near the target insertion site. This insertion completed the selected half sites and allowed the endonuclease digestion of intermediate bacmids, removing the resistance cassette. The resulting linearized vector was then re-circularized with insert fragments carrying the target transgene and adjacent removed viral regions. This combination was conducted via Gibson assembly [16].

A detailed description of all cloning processes is provided in Appendix B.

Appendix A lists all plasmids and bacmids used in and for this study, along with their features and accession numbers. Appendix A lists the primers used for the amplification of bacmid backbones designated for AdV genome assembly. The remaining primers are listed in Appendix A. Appendix A lists the synthetic DNA fragments used for the assembly of the transfer vectors.

### 2.3. Rescue of Recombinant Adenovirus

The viruses were rescued as described previously [14]. Briefly, 293A, A549, or HeLa cells were co-transfected with viral bacmids and a plasmid coding for Cas9-nuclease and sgRNA, targeting bacmid sequences adjacent to the viral ITRs. The binding of this sgRNA allows Cas9 to induce double-strand breaks, thereby releasing the viral DNA within the transfected cells. As a helper plasmid, we used pAR-Cas9-Ex [14], which was designed to express sgRNA-Ex and bind to an artificial CRISPR/Cas9 target (ACT) site adjacent to the ITR. As ITRs vary among different AdV types, this allowed using a common helper plasmid for all virus types rather than having to modify the sgRNA depending on the target virus. But, beside this generally applicable sgRNA, other targeting sgRNAs recognizing the viral ITR–bacmid junctions could also be designed to rescue HAdV-C5- and -E4-based recombinants, named as Int5 and Int4. To test the applicability of Int-targeting sgRNAs, sgRNA-Int4 was used for the rescue of HAdV-B3 and HAdV-E4 and sgRNA-Int5 for HAdV-B35 and HAdV-F41, as the ends of the ITR sequences are shared between these types.

The co-transfected cell were sub-cultured 24 h after co-transfection and observed daily for plaque formation. After the emergence of CPE, the cells were harvested and lysed via serial freeze–thaw cycles, releasing progeny viruses, which could then be expanded using serial passaging. In the case that plaque formation was not evident until day 8 of post-transfection, the transfected cells were lysed by freeze-thawing, and the sub-confluent cells were treated with 200 μL/10 cm^2^ of these lysates and observed for another 7 days for plaque formation (blind passage).

### 2.4. Time Course Analysis

Growth curves were generated to investigate the growth characteristics of recombinant viruses. A549 cells were infected at MOI 0.1 in DMEM, lacking FCS and P/S. After 90 min, the infection medium was removed, the cells were washed with PBS, and fresh DMEM supplemented with 2% FCS and 100 U/mL P/S was added. After this, samples were taken at 0, 6, 12, 24, 36, 48, 60, and 72 h post-infection (hpi). The samples were taken by harvesting the cells and supernatant and were immediately frozen at −80 °C. To determine the transgene expression, the samples were thawed, and 20 μL supernatant was mixed with 50 μL luciferase reagent. Luminescence was detected 10 min later using TECAN Infinite M-Plex. The samples were then re-frozen, re-thawed twice more, and used in serial dilutions from 1 × 10^−2^ to 1 × 10^−9^ to infect fresh A549 cells. Each sample was titrated in 4 separate dilution series in parallel. Infection could be observed seven days later using a KEYENCE fluorescent microscope for transgenic viruses. Counting the infected wells as determined by fluorescent foci formation could then allow for calculating TCID50 and PFU, as described previously [18]. For wild-type and E3-deleted virus, infection was determined using an adjusted AAV-based replicon assay variant [19]. Nine days post-infection, 293A cells transduced via AAV-expressing Gaussia luciferase were added to each well. One day later, 20 μL supernatant was mixed with 50 μL luciferase reagent, and luminescence was detected 10 min later using TECAN Infinite M-Plex. Luminescence that increased >100-fold over the non-infected control wells was used to determine positive infection, and TCID50/PFU was calculated as described above.

### 2.5. Sequencing and Data Analysis

Sanger sequencing was performed by Azenta. Next-generation Illumina sequencing was performed in house by the Clinical Virus Genomics unit. Data obtained from sequencing were analyzed as described previously [20].

Statistical significance was determined by an ordinary one-way ANOVA or a two-way ANOVA when applicable. Tukey’s multiple comparison test was used to analyze the significance between the mean values. Significance is indicated as follows: * *p* < 0.01, ** *p* < 0.001, and *** *p* < 0.0001.

### 2.6. Software

In silico sequence analysis and planning were performed using Geneious Prime 2019.2.3. [21]. NGS data were processed using a custom Galaxy (usegalaxy.eu; accessed on 12 December 2023) [22] workflow. Statistics were generated using GraphPad Prism 8.1. The figures were generated using GraphPad Prism 8.1, BioRender.com (accessed on 18 April 2024), and Affinity Designer 2.0.4.

## 3. Results

### 3.1. Screening of Mastadenovirus Types Usable for CTR

First, we wanted to test if CTR would be an applicable method to broadly rescue recombinant adenoviruses.

As the applicability of CTR was shown for HAdV-C5 and HAdV-E4 [14], we decided to test the following adenovirus types:Human adenovirus species A, type 12;Human adenovirus species B1, type 3;Human adenovirus species B2, type 35;Human adenovirus species C, type 2;Human adenovirus species F, type 41;Simian adenovirus species E, type 25.

To construct viral bacmids for types HAdV-A12, -B3, -B35, -C2, and -F41 and SAdV-E25, we applied a Gibson assembly-based methodology, which was first described by Pan et al. for constructing genomic high-copy plasmids [13] and modified for bacmid construction [14] (see the schematic in Figure 1a). In rescue experiments testing HAdV-C5 and E4, we used wild-type bacmids described previously [14]. As the assembly of viral genomes into bacmids is based around homologies between ITRs and bacmid fragments, viral DNA can be assembled facing both directions within the bacmid, either with bacterial replication genes and viral E1-region reading in the same (Figure 1b) or opposite directions, named in this paper reverse complement assembly (RC, Figure 1c).

Notably, some investigated bacmids were biased towards one assembly orientation, present in a majority or exclusively in all bacterial clones (see Table 2). While all bacmid clonings yielded hundreds of colonies, the efficiency of gaining correct constructs varied significantly between different viral genomes, as shown by the restriction analysis of ten randomly picked primary bacterial colonies.

Following the cloning of the viral bacmids, the virus was rescued by CTR, as described previously for HAdV-C5-based first-generation vectors using 293A cells. However, we expanded the target cell range, including HeLa and A549 cells, for rescue experiments for most bacmid constructs. While all experiments yielded progeny viruses at least after the transfection of one cell type, the rescue results varied among the cell lines and the AdV types (see Table 3). In cases where viral plaques were not readily apparent, the cells could be harvested eight days post-transfection and lysed as usual. When using these lysates to start serial passaging, plaques became apparent, indicating progeny viruses being present despite no initial plaque formation at these earlier time points.

Since all bacmids had the same ACT sequence, we utilized sgRNA-Ex to rescue all tested adenovirus types. As previously demonstrated, cutting proximally to the ITR significantly increases primary rescue efficiency, making the generation of viral libraries easier. We also wanted to test this approach for the novel types generated ins this paper and tried high-efficiency rescue using specific sgRNA-Int4. This sgRNA was designed to facilitate cutting directly at the end of the ITRs of HAdV-E4 (see Figure 2). Since HAdV-E4 and HAdV-B3 share the same sequence at the ends of their ITRs, sgRNA-Int4 could also be used to release HAdV-B3 proximally to the ITR from genomic bacmids. Similarly, sgRNA-Int5 was cross-compatible with HAdV-B35 and HAdV-F41. However, while the rescue of HAdV-E4 worked well and yielded more primary viral particles [14], no progeny virus could be observed for HAdV-B3 when we used sgRNA-Int4-expressing helper plasmid. Conversely, sgRNA-Int5-expressing helper plasmid was successfully used to rescue both HAdV-B35 and HAdV-F41.

Using the appropriate cell lines seemed to be an essential factor in facilitating virus rescue. While A549 allowed the rescue of all bacmids carrying the wild-type AdV genomes generated in this study, other cell lines allowed a higher primary yield and faster plaque observation or did not permit rescue. Table 3 provides an overview of the cell lines used and tested. Notably, 293A cells did not permit the rescue of HAdV-B3 but allowed the rescue of HadV-B35 to a similar degree to A549 cells.

### 3.2. HFR Allows for the Reliable Assembly of Novel Vectors

As we previously established HFR as a method for the rapid and highly efficient modification of adenoviral bacmids [12] based on HadV-C5, we wanted to test if this was universally applicable as we hoped. To test this, we chose to modify HadV-B3 and HadV-E4 as a basis for transgene insertion. HadV-B3 is often referenced as an up-and-coming candidate for oncolytic therapy due to its primary receptor usage of DSG-2 [7,23,24]. At the same time, HadV-E4 is attractive as both an oncolytic vector and a vector for vaccine applications. Its seroprevalence is relatively low worldwide [25], and it induces a strong and long-lasting immune response as a replicating mucosal vector [25,26]. In both cases, applications using replication-competent vectors are foreseen. Therefore, we applied HFR targeting multiple insertion sites along both genomes.

All transgenic variants were based on E3-deleted viral recombinants, as the packaging of oversized genomes is generally inefficient in adenoviruses, and E3 is non-essential for virus propagation in cell cultures. As HFR allows for the seamless insertion of modification at any place within the genome, we selected six different potential sites for insertion of the transgene (see Figure 3a; for the precise genomic location, see Appendix A). The sites were designed to avoid disrupting the known coding sequences and regulatory regions. Site I0n was targeted between the viral packaging domain and the start of the E1A region, while I0 utilized the intergenic region between E1A and E1B. I1 is located behind the coding regions of pIX and pIVa2. I2 is located within the former E3 region. I3 uses the intergenic region between the L5 and E4 regions. Lastly, I4 is located between the right ITR and the start of the E4 region. Insertion was performed via HFR, using PCR products of transfer pUC-based vectors carrying original viral sequences and dual-reporter cassette-expressing GFP and Gaussia Luciferase (GLuc), augmented with a chimeric intron (CI) cloned from pMT2 [27] for higher expression, schematically shown in Figure 3b. Due to the required size of PCR products, some recombinants were built using two overlapping PCR products of the same vector. All recombinant bacmids were successfully cloned, and assembly was confirmed using RFLP, Sanger sequencing, and Illumina sequencing.

The rescue was performed as outlined before; however, neither Insertion sites I0 nor I0n yielded any progeny virus, even after the lysis of the transfected producer cells. Other insertion sites were successfully rescued for both HAdV-B3 and HAdV-E4.

### 3.3. Insertion Site Directly Affects Transgene Expression but Does Not Affect Viral Replication

Viral stocks of vectorized viruses were successfully generated and titrated. Over the course of three days, we investigated the potency of chosen insertion sites and how this might affect viral fitness, both transgene expression and viral replication.

A549 cells were infected at MOI 0.1, and samples for expression testing and growth kinetics were taken at indicated time points. As GLuc is secreted from infected cells, we could check transgene expression via the supernatant (Figure 4a,b). Notably, I4 displayed significantly reduced transgenic expression compared to other insertion sites for both viruses, as determined by measuring the area under the curve (AUC) of luciferase activity (Figure 4c,d). In HAdV-B3, the insertion sites I1, I2, and I3 display similar expression levels and kinetic levels, reaching a plateau at ~36 hpi. At 72 hpi, expression is universally decreased slightly for these insertion sites, while I4 remains stable, while having a lower overall expression. HAdV-E4-derived recombinants behave similarly. However, a plateau is less apparent here. Notably, only I2 seems to provide a further increase in expression within the measured timeframe (Appendix A), while only I1 displays a dip in expression at 72 hpi observed for HAdV-B3. The overall expression of I1 is also decreased. However, this difference is not statistically significant (E4-I1 vs. E4-I2: *p* = 0.1052; E4-I1 vs. E4-I3: *p* = 0.1072). These differences also became apparent when comparing viral plaques using fluorescent microscopy (Figure 5a). To investigate if this effect might be linked to an inhibition or increase in viral replication, we performed a growth curve analysis on the lysates obtained from these experiments (Figure 5b). Notably, no highly significant differences in viral variants could be observed, and no impairment was observed compared to wild-type or E3-deleted virus variants.

## 4. Discussion

In this work, we aimed to investigate a reverse genetic workflow for modifying adenoviral bacmids and their rescue into progeny viruses throughout most adenovirus species. Using a selection of types representing different human and a simian adenovirus species, we tested our bacmid cloning approach based on the Gibson assembly-based method by Pan et al. [13]. This way of cloning appeared applicable for the construction of all tested AdV genomes and allowed the easy establishment of CTR-permissive bacmid vectors based on pKSB2 [17]. Interestingly, some bacmid assemblies revealed a bias for an orientation of the genomic insert. Gibson assembly could also permit assembly in reverse complement, yet fewer or no clones could be observed carrying specific configurations. The preferred orientation of the genome seemed to vary among the different types of adenovirus genomes, with no pattern revealing the relation to virus types readily apparent. We hypothesize that this could be due to the emergence of new ORFs, coding potentially toxic products in *E. coli*; however, the in silico analysis for larger fusion ORFs spanning the bacmid vector–adenovirus borders did not yield conclusive results.

While CTR was likewise widely applicable, its results were not uniformly satisfactory. If novel adenoviral bacmids are to be rescued by CTR, one should take good care to test more permissive cell lines, as some might not permit the rescue of the specific virus. Notably, HAdV-B3 could not be rescued in 293A cells, which are the most commonly used cells in adenovirus production, primarily due to the trans-complementation effect of the HAdV-C5 E1-region, which allows the rescue and propagation of replication-deficient viral vectors of species C but also for most other types of human adenovirus vector platforms.

This restriction of HadV-B3 is unclear as DSG-2 is readily expressed on the surface, and 293A cells are generally permissive for HadV species B infection [28]. Interestingly, HadV-B35 rescued well in 293A cells. In previous experiments, we saw that, while HAdV-B3 can successfully infect 293A cells, the replication and expression of the viral proteins are significantly inhibited and delayed [29]. This effect is also worth considering when investigating E1-deleted viruses, as this is the current standard for HAdV-C5-based vectors [30,31]. 293A cells equipped with HAdV-C5-E1 can cross-complement other adenoviral types, such as HAdV-D64 [32] and HAdV-E4 [33], but this seems to differ for different species and types. Notably, previously generated E1-deficient HAdV species B viruses were not generated in 293A cells [34,35,36,37] (except for HAdV-B7 [38]), but rather other trans-complementing producer cells lines, such as Per.C6 [39] or 911 [40] cells. HAdV-B35, produced in this study, was successfully rescued from 293A cells. However, the overall efficiency was comparable to the rescue in A549 cells, which is surprising, as A549 cells are notoriously more difficult to transfect, further supporting our idea of 293A inhibiting species B viruses in some way. This aspect needs to be assessed carefully with specifically designed experiments for testing virus rescue quantitatively, which was not the focus of this study.

Overall, we see this as an indication that selecting a cell line exhibiting a high transfection efficiency and permissibility to viral replication is the primary factor for high-efficiency viral rescue, not only the selection of specific sgRNAs, as assumed previously [14]. Especially when considering library applications, which we presented previously for HAdV-C5 using 293A cells [12], selecting the appropriate alternative cell lines will be required for library applications to modify HAdV belonging to other species, such as B, E, and F.

Yet, ITR-proximal cuts (using different sgRNA-Int) are an important determinant of rescue efficiency. Proximal cuts always resulted in a higher primary yield than sgRNA-Ex-induced rescue when they were successfully applied. However, proximal cuts, unlike distal ones, were not universally applicable to the types investigated in this paper. Notably, HAdV-B3 again seems to suffer in this respect, with sgRNA-Int4-supplemented rescue attempts failing to yield any progeny virus. As the same sgRNA was used to efficiently rescue HAdV-E4 [14] and sgRNA-Ex successfully rescued the bacmid containing HAdV-B3, we conclude that this inhibition is a specific effect of gRNA-Int4. As previously discussed [14], gRNA-Ex could be designed in a way to warrant high efficiency with a high activity score in mind (0.715 of gRNA-Ex when targeting HAdV-E4), and specific gRNAs are limited in design freedom due to binding directly to the ITRs. gRNA-Int4 displays a far lower activity score of 0.067, potentially allowing for more off-target binding. Our current hypothesis is that off-target binding within the HAdV-B3 genome can occur due to this low specificity, inducing double-strand breaks and rendering genomic DNA unable to replicate, or it may target a host cell locus, which is irrelevant for HAdV-E4 replication but essential for HAdV-B3 permissivity. Translating previously established library approaches to these newly constructed HAdV types might not be universally applicable with specific gRNAs at the same scale and needs to be investigated on a per-species level.

As in our previous work, using HFR to generate single recombinants of different species proved highly reliable and efficient [12]. Despite the rational design of insertion sites targeting the inert regions of the genome, inserting a transgene before the E1 coding region or between E1A and E1B yielded no progeny virus. As the E1 region is essential to viral replication, it is possible that inserting larger transgenic cassettes at these sites disturbs transcription factor binding sites or other cis-acting elements in the area, e.g., the ITR [41,42]. While we tried to avoid the known regulatory sequences and packaging sequences, many of those are based on homologies to HAdV-C5 and, therefore, not fully understood in other species. As such, the insertion of transgenes within these highly volatile regions is not recommended for novel species.

Other investigated insertion sites, however, yielded viable progeny virus. Interestingly, all insertion sites located between the 3′ coding regions of their respective strands (I1, between pIX and antisense pIVa2; I3, between the L5-region and antisense E4-region) allowed expression similarly to I2, which was located in the classically replaced E3-region. The removal of E3 is highly desirable for transgenic applications, as the removal of E3 reduces the genome size, permitting the insertion of larger transgenes without surpassing the ideal genome size to warrant optimal packaging capabilities and virus stability [43,44]. However, for applications where E3 or parts of E3 should be retained, as it might improve target vector characteristics [45,46], I1 and I3 provide alternative insertion sites without impacting transgene expression. Nevertheless, it is worth noting that HAdV-E4 expression was slightly reduced for I1-based vectors, yet this effect was not statistically significant. The I2-based expression also showed steadily increasing expression; so, the viability of these insertion sites needs to be once again assessed for specific applications and vector candidates.

For both viruses, I4-based vectors failed to produce competitive levels of transgenes, always remaining significantly below the insertion sites mentioned above. We initially expected a growth defect to cause this, as all experiments were performed using a low initial virus load and investigated over several days. However, we learned that all viruses showcase similar growth, remaining at wild-type-like levels. As such, other factors, such as the accessibility of the viral genome being restricted (e.g., by DNA binding protein, which was previously shown to inhibit E4 activation [44]) or competing transcription factors [45], might cause this offset. However, directly impacting E4 transcription seems unlikely, as E4 has numerous functions important to viral replication, such as the transactivation of viral genes and binding to cellular factors [47,48,49,50,51]. As such, an impact on E4 function would most likely cause an impact on growth characteristics, which we could not observe in this study.

In summary, the insertion of transgenes near the viral promoter region significantly affects virus viability (I0n, I0) or inhibits transgene expression (I4). Other insertion sites between the tail-to-tail polyA sites of viral genes not only allow a high level of expression but also do not inhibit viral replication, making them highly attractive for use in replicating viral vectors.

Our data also suggest a mild, non-significant delay in replication for non-transgenic E3-deleted HAdV-E4, again showing the importance of proper genome size not impacting viral fitness, even if this effect is not statistically significant. In this specific use case, it is also worth pointing out the packaging capabilities of our replication-competent vectors. While removing E3 in HAdV-C5 frees up roughly 3 kbp of genomic space, removing E3 in HAdV-B3 and HAdV-E4 frees up ~3.8 kbp and ~4.4 kbp, respectively. This might become relevant, especially for larger transgenes both in clinic and research usage.

Previous works already established fast and efficient methods for building adenoviral bacmids and modifying them successfully [52]. However, in this work, we combined our novel approaches to further the applicability of this platform. Direct cloning into CTR-permissive bacmid allows for the rapid and immediate rescue of viral particles compared to conventional methods [14,53] while adding the benefit of library applicability, allowing for further applications such as in vitro evolution to select optimized viral variants [54,55,56,57]. With CTR and HFR being widely applicable among adenoviruses, we propose this as a platform for cloning single recombinants and generating genetic libraries, as demonstrated previously [12]. However, the careful optimization and selection of proper components is required to ensure the maximum fidelity and efficiency of this workflow when applied to novel HAdV types.

While only shown in this paper as a tool for inserting transgenes, HFR also allows for the easy and seamless modification of viral genes. Especially when used for multiple mutants of the same gene, the common intermediate precursor bacmid allows for rapid turnover and virus generation when paired with HFR.

## Figures and Tables

**Figure 1 viruses-16-00658-f001:**
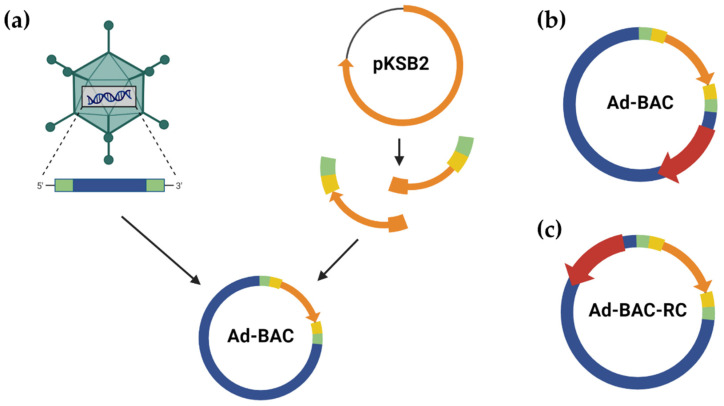
Assembly of bacmids carrying AdV genomic DNA. (**a**) Schematic overview of bacmid assembly, starting with genomic DNA (blue) extraction from viral particles. The plasmid backbone (orange arrow) is amplified from pKSB2 via PCR, adding ACT sequences (yellow boxes) and homologies to viral ITR (green) as well as internal overlaps between PCR fragments (orange boxes). Fragments are combined with DNA using Gibson assembly. The finished assembly products can incorporate genomic DNA in 2 orientations, either (**b**) with E1A (red arrow) and backbone coding in the same direction or (**c**) with E1A and backbone coding in opposite directions.

**Figure 2 viruses-16-00658-f002:**
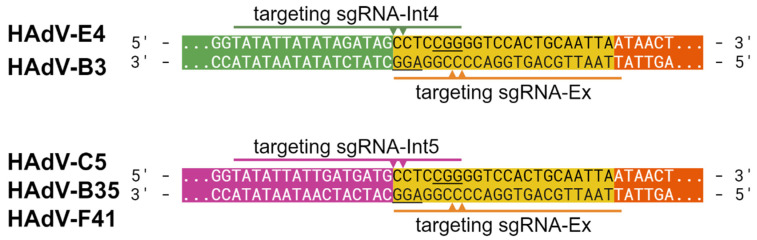
sgRNAs used for CTR. Schematic view of ACT sequences (yellow) and adjacent ITR (green/magenta) and bacmid (orange) sequences present on genomic bacmids carrying the indicated HAdV genomes. The targeted sites of the generally applicable (sgRNA-Ex) and the more specific (sgRNA-Int4 and -Int5) sgRNAs used for CTR are shown below or above the sequences, respectively, with arrows indicating cutting sites. PAMs are underlined.

**Figure 3 viruses-16-00658-f003:**
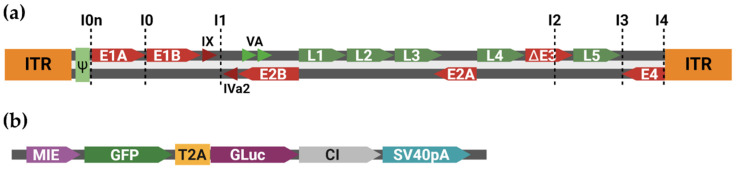
Generation of replication-competent reporter virus vectors. (**a**) Schematic view of HAdV genome, with the target sites for transgenic insertion indicated. Early genetic regions are marked red, late genetic regions in green. Transgenic viruses were based on E3-deleted variants to make appropriate space for transgene cassette; (**b**) Schematic view of inserted transgene cassette; MIE: MCMV immediate early promoter; GFP: Green fluorescent protein; GLuc: Gaussia luciferase; CI: Chimeric intron cloned from pMT2; SV40pA: Poly-A tail of SV40.

**Figure 4 viruses-16-00658-f004:**
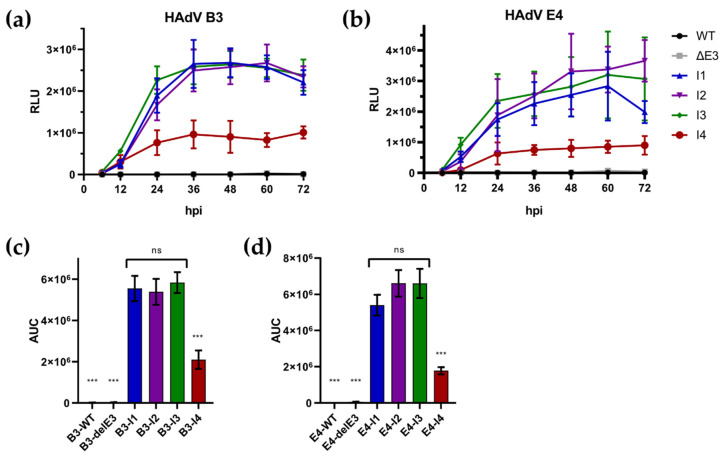
Expression of transgenes over time as determined by a Luciferase assay for (**a**) HAdV-B3-based recombinants and (**b**) HAdV-E4-based recombinants. Area under the curve for luciferase measurements of (**a**,**b**), displayed for easier comparison for (**c**) HAdV-B3-based recombinants and (**d**) HAdV-E4-based recombinants. Statistical significance was determined by an ordinary one-way ANOVA with Tukeys’s multiple comparisons. Significance is indicated as follow: *** *p* < 0.0001, and ns: not significant.

**Figure 5 viruses-16-00658-f005:**
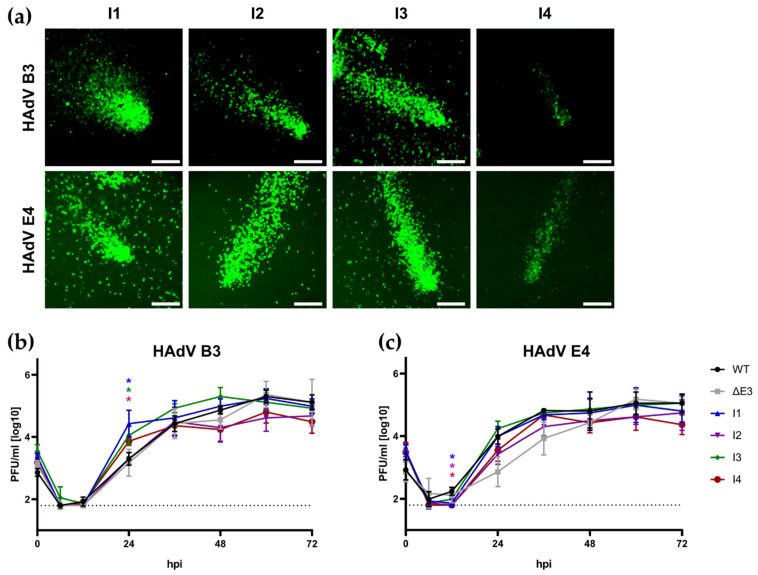
(**a**) Fluorescent microscopy of representative plaques of different recombinant viruses. Contrast is boosted equally across all pictures for better clarity. Scale bar represents 200 μm. Growth curves for recombinant viruses compared to wild-type and non-transgenic E3-deleted construct for (**b**) HAdV-B3 and (**c**) HAdV-E4. Dashed line represents lower detection limit. Statistical significance was determined against the corresponding wild-type virus. Statistical significance was determined by an ordinary two-way ANOVA. Significance is indicated as follow: * *p* < 0.01.

**Table 1 viruses-16-00658-t001:** Overview of the viruses used in this study.

Virus	Source
HAdV-A12	Prof. Dr. Anja Erhardt, University Witten-Herdecke
HAdV-B3	Prof. Dr. Anja Erhardt, University Witten-Herdecke
HAdV-B35	Prof. Dr. Albert Heim, German Adenovirus Reference Laboratory, Hannover
HAdV-C2	ATCC VR-846
HAdV-E4	ATCC-1572
HAdV-F41	ATCC VR-930
SAdV-E25	ATCC VR-594

**Table 2 viruses-16-00658-t002:** Overview of the bacmid assemblies of genomic AdV DNA. Clones showcasing the correct restriction pattern were analyzed for the orientation of genomic DNA.

	Human							Simian
Species	A	B1	B2	C		E	F	E
Type	12	3	35	2	5	4	41	25
Ad-BAC	-	3	-	-	7	2	-	4
Ad-BAC-RC	3	2	9	8	2	1	4	2
Correct clones *	30%	50%	90%	80%	90%	30%	40%	60%

* as determined by restriction pattern analysis.

**Table 3 viruses-16-00658-t003:** Overview of the cell lines used for virus rescue and their relative permissiveness for virus reconstitution. +++: High plaque counts, visible ≤ 4 days after transfection; ++: Plaques visible 5–8 days after transfection; +: Plaques visible > 8 days after transfection or blind passaging used to recover infectious particle; −: No successful rescue; ND: Not determined.

	Human							Simian
Species	A	B1	B2	C		E	F	E
Type	12	3	35	2	5	4	41	25
HeLa	ND	++	++	ND	++	++	−	ND
A549	++	++	+++	++	++	++	+	+
293A	+++	−	+++	+++	+++	++	+	+

## Data Availability

Data presented in this study are available from the corresponding author upon reasonable request.

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
