# Peer review of "Expanding the Scope of Adenoviral Vectors by Utilizing Novel Tools for Recombination and Vector Rescue"

_viruses, 2024, doi:10.3390/v16050658_

Round 1
Reviewer 1 Report
Comments and Suggestions for Authors
Adenoviruses (AdV) have been studied for many years as a model for virus-host interactions and numerous seminal discoveries in virology and cell biology have been described. A wide variety of human AdV have been identified that can lead to inapparent infection or mild disease, but severe disease may occur under some circumstances. AdV have been widely used as vectors for gene expression in vitro and in vivo and in many different types of human gene therapy approaches. In this report, the authors describe approaches to readily clone and manipulate the genomes of different human AdV using Bacmids and Gibson assembly and CRISPR-Cas9 to excise viral genomes from Bacmids. They demonstrate the feasibility for all seven human AdV species, characterize the growth if recombinant AdV in different human cell lines, and analyze sites fo insertion in the non-essential E3 region and how these affect transgene expression. This is a worthwhile manuscript to the field and the results are technically clean and clearly explained.
Minor point: Title of section 3.3 uses the word "affection" which makes no sense in this context. I assume they mean "expression."
Describe a generalized workflow that allows vectorization, res-14 cue, and cloning of all adenoviral species to enable the rapid development of new vector variants. 15 We show this on human and simian adenoviruses, further modifying a selection of these to investi-16 gate their gene transfer potential and build potential vector candidates for future applications.
Author Response
Dear Reviewer 1,
We thank you for your thorough and vigilant review of our manuscript and thank you for your comment.
You are correct in your assumption about the title of section 3.3, we corrected the text to correctly read "expression" (Page 7, line 275).
Thank you once again and best wishes,
Julian Fischer and Zsolt Ruzsics
Reviewer 2 Report
Comments and Suggestions for Authors
Authors have previously reported the use of a novel recombinant adenovirus construction method for human Ad5 and Ad4. In this paper this method is explained in detail an applied to six other adenovirus types, five human and one simian.
The method applies the most advanced molecular biology techniques and it is efficient.
Virus genomes are first incorporated into a bacmid using Gibson assembly recombination. In this bacmid an artificial CRISPR-Cas target is cloned flanking the ITRs.
Genome modifications (with donor fragments) are performed by inserting endonuclease sites by homologous recombination and later using those sites to open the bacmid for Gibson assembly with donor fragments.
Genome release upon transfection using CRISPR/Cas and guides that target it to the sites flaking the ITRs.
Only minor comments:
Whati is CI in fig 3b?
Regarding the selection of the half-sites of endonucleases for the targeted insertions, although appendix A mention the sites, may be figures further explanations (as with Swa in their previous work) would be useful.
Page 8, line 280 mentions: “As GLuc is sequestered from infected cells, we could check transgene expression via the supernatant (Figure 4a/b)”. It is not clear to me: if GLuc is not secreted (it is sequestered) why can be detected in the supernatant?
Author Response
Dear Reviewer 2,
We thank you for your thorough and vigilant review of our manuscript and thank you for your comments.
We added a short description of "CI" to the legend of figure 3b and added additional information and a reference to the main text (Page 7, lines 266/267).
To illustrate HFR used for cloning in this study a bit better, we added a new figure to the appendix (page 13), showcasing the cloning and selection process on the example of Insertion site I1. We also added links to the figure at the appropriate locations in the text (page 13, lines 479,483,499,504).
You are correct in your assessment that sequestration does not make sense in the context. We adjusted the text to correctly read "secretion".
We thank you once again and best wishes,
Julian Fischer and Zsolt Ruzsics